# Pneumococcal colonization prevalence and density among Thai children with severe pneumonia and community controls

**Barameht Piralam[1¤a]\***, **Christine Prosperi[2☉]**, **Somsak Thamthitiwat[3]**, **Charatdao Bunthi[3]**, **Pongpun Sawatwong[3]**, **Ornuma Sangwichian[3]**, **Melissa M. Higdon[2‡]**, **Nora L. Watson[4‡]**, **Maria Deloria Knoll[2‡]**, **Wantana Paveenkittiporn[5]**, **Chuwattana Chara[6]**, **Cameron P. Hurst[1¤b]**, **Pasakorn Akarasewi[7]**, **Julia Rhodes[3¤c]**, **Susan A. Maloney[3¤d]**, **Katherine L. O'Brien[2¤e]**, **Henry C. Baggett[3☉¤f]**

1 Department of Epidemiology and Biostatistics, Khon Kaen University, Khon Kaen, Thailand, 2 Department of International Health, International Vaccine Access Center, Johns Hopkins Bloomberg School of Public Health, Baltimore, Maryland, United States of America, 3 Division of Global Health Protection, Thailand Ministry of Public Health–US Centers for Disease Control and Prevention Collaboration, Nonthaburi, Thailand, 4 The Emmes Company, Rockville, Maryland, United States of America, 5 Ministry of Public Health, National Institute of Health, Nonthaburi, Thailand, 6 Nakhon Phanom Provincial Hospital, Nakhon Phanom, Thailand, 7 Department of Disease Control, Ministry of Public Health, Nonthaburi, Thailand

☉ These authors contributed equally to this work.
¤a Current address: Nakhon Phanom Provincial Health Office, Nakhon Phanom, Thailand
¤b Current address: QIMR Berghofer Medical Research Institute, Herston, Queensland, Australia
¤c Current address: National Center for Immunization and Respiratory Diseases, Centers for Disease Control and Prevention, Atlanta, Georgia, United States of America
¤d Current address: Division of Global HIV and TB, Center for Global Health, Centers for Disease Control and Prevention, Atlanta, Georgia, United States of America
¤e Current address: World Health Organization, Initiative for Vaccine Research, Geneva, Switzerland
¤f Current address: Division of Global Health Protection, Centers for Disease Control and Prevention, Atlanta, Georgia, United States of America
‡ These authors also contributed equally to this work.
\* baramehtp@ieipnp.go.th, baramehtp@gmail.com

**Data Availability Statement:** Due to legal and ethical restrictions prohibiting public sharing of a data set, the data are not yet publicly available.

## Abstract

### Background

Pneumococcal colonization prevalence and colonization density, which has been associated with invasive disease, can offer insight into local pneumococcal ecology and help inform vaccine policy discussions.

### Methods

The Pneumonia Etiology Research for Child Health Project (PERCH), a multi-country case-control study, evaluated the etiology of hospitalized cases of severe and very severe pneumonia among children aged 1–59 months. The PERCH Thailand site enrolled children during January 2012–February 2014. We determined pneumococcal colonization prevalence and density, and serotype distribution of colonizing isolates.

PERCH developed an Agreement for Data, Dissemination, and Specimens, signed by all PERCH institutions in April 2016 which specifies the procedures. This requires external investigators to submit a request to the PERCH study team explaining the proposed use of the data then signing a Confidential Disclosure Agreement before de-identified data are shared (see Annex A1, starting on Page 18). Making the data publicly available at this time would violate the current Agreement in place with all PERCH institutions. We are working with the institutions to revise the Agreement in parallel with setting up the data repository in ClinEpiDB (University of Pennsylvania). We anticipate the Agreement will be revised and the data made available by the end of Q3 2020. In the meantime, researchers can access the data by contacting a representative on the PERCH Core Team (Amanda Driscoll, ADriscoll@som.umaryland.edu or Christine Prosperi, cprospe1@jhu.edu). Researchers will need to complete a dataset access request form explaining their planned analysis which will be reviewed by the PERCH Executive Committee. After signing the PERCH Confidential Disclosure Agreement the data will be made available to the researcher.

**Funding:** Funding Statement and Competing Interests Statement This work was supported by the Bill & Melinda Gates Foundation (grant 48968 to the International Vaccine Access Center, Department of International Health, Johns Hopkins Bloomberg School of Public Health, for the PERCH study). Representatives from the Bill & Melinda Gates Foundation participated in site selection and in Pneumonia Methods Working Group meetings, which informed the study design. They had no role in the data collection, data analysis, data interpretation, or writing of the report. The funder provided support in the form of salaries for some co-authors (B.P., C.C., C.P., M.M.H., M.D.K., and K. L.O.), but did not have any additional role in the study design, data collection and analysis, decision to publish, or preparation of the manuscript. The specific roles of these authors are articulated in the 'author contributions' section.

**Competing interests:** N.L.W., who was involved with analyses and interpretation of results, was employed by a commercial data management and statistical analysis company, The Emmes Corporation. This does not alter our adherence to PLOS ONE policies on sharing data and materials. There are additional potential competing interests to note among co-authors: M.D.K. has received funding for consultancies from Merck, Pfizer, Novartis, and grant funding from Merck. M.M.H.

## Results

We enrolled 224 severe/very severe pneumonia cases and 659 community controls in Thailand. Compared to controls, cases had lower colonization prevalence (54.5% vs. 62.5%, p = 0.12) and lower median colonization density (42.1 vs. 210.2 x $10^3$ copies/mL, p <0.0001); 42% of cases had documented antibiotic pretreatment vs. 0.8% of controls. In no sub-group of assessed cases did pneumococcal colonization density exceed the median for controls, including cases with no prior antibiotics (63.9x$10^3$ copies/mL), with consolidation on chest x-ray (76.5x$10^3$ copies/mL) or with pneumococcus detected in whole blood by PCR (9.3x$10^3$ copies/mL). Serotype distribution was similar among cases and controls, and a high percentage of colonizing isolates from cases and controls were serotypes included in PCV10 (70.0% and 61.8%, respectively) and PCV13 (76.7% and 67.9%, respectively).

## Conclusions

Pneumococcal colonization is common among children aged <5 years in Thailand. However, colonization density was not higher among children with severe pneumonia compared to controls. These results can inform discussions about PCV introduction and provide baseline data to monitor PCV impact after introduction in Thailand.

## Introduction

*Streptococcus pneumoniae* (pneumococcus) is a leading cause of bacterial pneumonia, meningitis, and sepsis in children worldwide [1, 2] and caused an estimated 294,000 (192,000–366,000) deaths globally in children less than 5 years in 2015 [2]. Pneumococcal colonization at the individual level, although usually asymptomatic, is a necessary precursor to invasive pneumococcal disease [3, 4] and an important source of spread in the community [4]. Young children generally have high colonization prevalence and serve as the primary reservoir for pneumococcus [5].

Prevalence of pneumococcal colonization is usually much higher in low- and middle-income countries [6], and the first acquisition happens at a younger age compared with high-income settings [3]. Although detecting pneumococcus in the nasopharynx or oropharynx of a child with pneumonia does not indicate an etiologic role, the density of colonization has been associated with pneumococcal disease [7–11], and while of uncertain value for individual diagnosis, may help to improve estimates of pneumococcal pneumonia prevalence in surveillance or observational studies. However, colonization prevalence and density vary by geographic location [7–11], so local data are needed to understand local epidemiology. Further, because pneumococcal conjugate vaccine (PCV) is serotype-specific, understanding the serotype distribution among colonized individuals informs estimates of potential vaccine effectiveness and is useful to monitor the impact of vaccine when introduced [3].

The Pneumonia Etiology for Child Health (PERCH) study was a multi-country case-control evaluation of the etiologic agents causing severe and very severe pneumonia among children in low- and middle-income countries in Africa and Asia [12]. A previous analysis of PERCH data from all study sites found an association between pneumococcal colonization density and microbiologically-confirmed pneumococcal pneumonia [7]. Country and regional differences in colonization density [7] warrant more detailed analysis of the Thailand data. While there are a few studies of pneumococcal colonization in Thailand [13–15] and other

has received grant funding from Pfizer. C.P. has received grant funding from Merck. K.L.O. has received grant funding from GlaxoSmithKline and Pfizer and participates on technical advisory boards for Merck, SanofiPasteur, PATH, Affinivax, and ClearPath.

Southeast Asian countries [16–22], more data are needed in well-characterized populations to describe trends in prevalence and serotype distribution [6]. Using data from the PERCH Study in Thailand, we assessed pneumococcal colonization density among young children hospitalized with pneumonia and its potential association with characteristics of pneumococcal pneumonia, while also describing colonization prevalence and serotype distribution among both ill and healthy children.

## Materials and methods

### PERCH project

The Pneumonia Etiology Research for Child Health (PERCH) was a multi-center case-control study of the etiologic agents causing severe and very severe pneumonia among children in seven low- and middle-income countries (nine study sites) [12].

### Study period, case and control definitions

PERCH enrollment in Thailand occurred in two study sites during January 2012 –December 2013 (Nakhon Phanom) and March 2012—February 2014 (Sa Kaeo). Identification of cases and controls has been described previously [23]. Cases were hospitalized children aged 1–59 months with World Health Organization (WHO)-defined severe or very severe pneumonia (using pre-2013 definitions) [24]. Cases were excluded if they had been hospitalized within the previous 14 days, had been discharged as a PERCH case within the previous 30 days, did not reside in the study catchment area, or had resolution of lower chest wall indrawing following bronchodilator therapy (for severe cases with wheeze).

Controls were randomly selected from the same community as cases and were frequency matched by the following age groups: 1 to <6 months, 6 to <12 months, 12 to <24 months, and 24–59 months [23]. Controls with respiratory tract illness (RTI) were included as long as they did not have severe or very severe pneumonia. RTI was defined as a presence of cough, runny nose, or if a child had (1) at least 1 of ear discharge, wheezing, or difficulty breathing, and (2) either a measured temperature of > 38.0˚C within the previous 48 hours or a history of sore throat. PERCH included controls with RTI to minimize bias in the comparison group for estimating pneumonia etiology [25].

### Specimen collection and laboratory testing

The standardization of laboratory methods for the PERCH study was described previously [26]. In summary, a flocked nasopharyngeal (NP) swab (flexible minitip, Copan®) and a rayon oropharyngeal (OP) swab specimen were collected from each case and control; the NP and OP swabs from each child were placed into the same vial with 3 ml of universal transport media (UTM) (Copan®). NP/OP specimens were kept at 4–8˚C for <24 hours and then stored at -70˚C until testing for pneumococcus (*lyt*A gene target) as part of a multiplex quantitative real-time polymerase chain reaction (PCR) assay (FTD Respiratory Pathogens 33, Fast-track Diagnostics, Sliema, Malta) performed using an Applied Biosystems 7500® (ABI-7500) platform (Applied Biosystems, Foster City, CA). Quantification of pneumococcal DNA as a measure of colonization density was performed using plasmid standards provided by Fast-track Diagnostics as described previously [7]. A second NP specimen for *S. pneumoniae* culture was collected simultaneously with the first swab, placed into a vial with 1 mL skim milk, tryptone, glucose, and glycerin (STGG) broth, kept at 4–8˚C for <24 hours and then frozen at -70˚C until cultured according to the WHO reference method using a broth enrichment step [27]. Serotyping of pneumococcal isolates was performed by multiplex PCR followed by

Quellung when necessary to resolve serotypes using previously published protocols [28]. Specimens that were positive for pneumococcus by PCR but not by culture were not serotyped. Blood from cases and controls was tested for *S. pneumoniae* by PCR and cultured for cases only. Serum was collected from cases and controls and tested for antimicrobial activity by serum bioassay: placing serum on a filter paper disc and plating a susceptible strain of *Staphylococcus aureus* and measuring growth inhibition around the inoculated disc [29]. Antibiotic pretreatment was defined as having either a positive serum bioassay or documentation of antibiotics administered at the study hospital prior to specimen collection. C-reactive protein (CRP) was assessed in all cases and in a subset of controls who were positive for pneumococcus by whole-blood PCR, had RTI, or were in the top 25% for total NP/OP PCR pathogen load (across all pathogens tested for).

### Clinical characteristics and outcomes

Chest radiographs (CXRs) were performed at admission for cases, and each digital image was assessed by two members of a panel of 14 radiologists and pediatricians who were trained in the standardized interpretation of pediatric CXRs [30, 31]. Clinical characteristics, including oxygen saturation, were assessed on the day of enrollment. Case mortality was assessed at hospital discharge and by contact at 30-day post discharge.

### Statistical analysis

Data were single-data entered into a centralised electronic data capture system (Emmes Corporation, Rockville, MD, USA) [32]. Pneumococcal colonization prevalence; demographic, clinical, and laboratory characteristics; and pneumococcal serotype distribution were compared between cases and controls using binary logistic regression adjusted for age. The Kruskal-Wallis test was used to compare the median pneumococcal colonization density across case and control subgroups. Additionally, we determined the percentage of colonizing isolates that were of serotypes included in PCV10 and PCV13 [33]. Serotypes 6A/6B by PCR could not be differentiated for the samples collected in 2013, so we applied the 6B:6A ratio from 2012 to the 2013 data. This calculation was done separately by site and case-control status. Unresolved serotyping data for other serogroups were left unresolved in analyses. Wald confidence intervals (95%) were calculated for the pneumococcal serotype distribution among cases and controls. Statistical significance was defined as $p < 0.05$. Statistical analyses were performed using SAS 9.4 (SAS Institute, Cary, NC, USA).

### Ethical considerations

The study was approved by the Institutional Review Boards (IRB) of the Johns Hopkins Bloomberg School of Public Health (protocol #3075), an IRB of the U.S. Centers for Disease Control and Prevention (protocol #6067), and the Thailand Ministry of Public Health Ethical Review Committee (protocol #17/2554). Parents or legal guardians of participants provided written consent for enrollment.

### Results

During the two-year study period, we enrolled 224 cases presenting to hospital with severe (n = 172, 76.8%) or very severe pneumonia (n = 52, 23.2%); two were excluded from this analysis because of missing NP/OP PCR results. Among 659 controls, 254 (38.5%) had RTI (Table 1); 650 controls had PCR results and were included in this analysis. **Antibiotic use before specimen collection was documented in 42.0% of cases and 0.8% of controls; an additional**

**Table 1. Demographic and clinical characteristics by case and control group in rural Thailand.**

| Demographic and clinical characteristics | [1] All cases (n = 222) | [2] All controls (n = 650) | [3] RTI controls (n = 250) | [4] Non-RTI controls (n = 400) | All cases [1] vs. all controls [2] p-value[a] | All cases [1] vs. RTI controls [3] p-value[a] | All cases [1] vs. non-RTI controls [4] p-value[a] | RTI controls [3] vs. non-RTI controls 4]] p-value[a] |
|---|---|---|---|---|---|---|---|---|
| **Provincial site** | | | | | | | | |
| Nakhon Phanom | 145 (65.3) | 429 (66.0) | 160 (64.0) | 269 (67.3) | 0.91 | 0.74 | 0.66 | 0.38 |
| Sa Kaeo | 77 (34.7) | 221 (34.0) | 90 (36.0) | 131 (32.8) | | | | |
| **Median age, months (IQR)** | 14.5 (7–26) | 16 (8–33) | 16 (9–34) | 16 (8–30) | **0.030** | **0.013** | 0.12 | 0.29 |
| **Age, n (%)** | | | | | | | | |
| 1–5 months | 38 (17.1) | 90 (13.8) | 25 (10.0) | 65 (16.2) | 0.69 | 0.16 | 0.96 | 0.14 |
| 6–11 months | 50 (22.5) | 150 (23.1) | 61 (24.4) | 89 (22.2) | | | | |
| 12–23 months | 68 (30.6) | 212 (32.6) | 81 (32.4) | 131 (32.8) | | | | |
| 24–59 months | 66 (29.7) | 198 (30.5) | 83 (33.2) | 115 (28.8) | | | | |
| **Male Gender** | 134 (60.4) | 330 (50.8) | 126 (50.4) | 204 (51.0) | 0.014 | 0.037 | 0.026 | 0.87 |
| **Prior antibiotic use, n (%)** | | | | | | | | |
| Any documented antibiotic pretreatment prior to specimen collection[b] | 94 (42.3) [c] | 5 (0.8) | 5 (2.0) | 0 (0.0) | < .0001 | < .0001 | < .0001 | < .0001 |
| Parental report only | 51 (23.0) | 51 (7.8) | 38 (15.2) | 13 (3.2) | | | | |
| No evidence of antibiotic use | 77 (34.7) | 594 (91.4) | 207 (82.8) | 387 (96.8) | | | | |
| **Fever[d]** | 123 (55.4) | 3 (0.5) | 3 (1.3) | 0 (0.0) | < .0001 | < .0001 | -- | -- |
| **Observed cough** | 189 (85.1) | 110 (16.9) | 110 (44.0) | 0 (0.0) | < .0001 | < .0001 | -- | -- |
| ***S. pneumoniae* in the NP/OP** | | | | | | | | |
| PCR Positive | 121 (54.5) | 406 (62.5) | 172 (68.8) | 234 (58.5) | 0.28 | **0.04** | 0.85 | **0.009** |
| PCR above threshold[e] | 3 (1.4) | 8 (1.2) | 7 (2.8) | 1 (0.3) | 0.61 | 0.61 | 0.054 | **0.023** |
| Culture positive | 89 (40.1) | 340 (52.4) | 155 (62.0) | 185 (46.4) | 0.07 | **0.002** | 0.77 | **0.0002** |
| PCR or culture | 127 (57.2) | 417 (64.2) | 177 (70.8) | 240 (60.0) | 0.31 | **0.043** | 0.87 | **0.008** |
| ***S. pneumoniae* in the whole blood** | | | | | | | | |
| PCR positive | 3 (1.4) | 5 (0.8) | 4 (1.7) | 1 (0.3) | 0.69 | 0.83 | 0.22 | 0.12 |
| PCR above threshold[f] | 0 (0.0) | 3 (0.5) | 2 (0.8) | 1 (0.3) | -- | -- | -- | 0.50 |

RTI, Respiratory tract illness; IQR, Inter quartile range. Table restricted to children with available nasopharyngeal/oropharyngeal (NP/OP) PCR results.

[a]P-values from logistic regression for categorical variables (adjusted for age in months for non-age variables) and Kruskal-Wallis tests for continuous variables. Comparisons of pneumococcus in the NP/OP and whole blood are also adjusted for antibiotic use. **Bolded p-values < 0.05.**

[b] Presence of antibiotics by serum, antibiotics at the referral hospital, clinician report of antibiotics prior to specimen collection or antibiotics prior to NP specimen collection based on time of specimen collection and time of antibiotic administration. Only criterion applicable to controls is serum. P-values calculated excluding those with parental report only.

[c]Prior antibiotic use (documented) stratified among cases stratified by site: 51.9% in Sa Kaeo and 37.2% in Nakhon Phanom, p = 0.03.

[d]Fever defined as measured temperature ≥ 38.0° C. Controls without signs or symptoms of illness in the past 48 hours were assumed to be without fever.

[e]NP/OP PCR density above 6.9 log10 copies/mL.

[f]WB PCR density above 2.2 log10 copies/mL.

**22.8% of cases (7.9% of controls) had received antibiotics by parental report only.** Cases in Sa Kaeo were more likely to receive antibiotics before NP/OP specimen collection than those in Nakhon Phanom (51.9% vs. 36.7%, p = 0.03). Two cases but no controls had received PCV.

*S. pneumoniae* was detected by PCR in the NP/OP of 121 (54.5%) cases, with no difference by pneumonia severity (p = 0.27), site (p = 0.39), CXR positivity (p = 0.23), or antibiotic use

before sampling (p = 0.92; Table 2). Colonization prevalence was not significantly different between cases and controls *(*p = 0.12). Controls with RTI had higher pneumococcal colonization prevalence (68.8%) than those without (58.5%, p = 0.01; S1 Table). Of 9 fatal cases, 5 (55.6%) received antibiotics before NP/OP collection; only one fatal case was colonized with pneumococcus, and this child received antibiotics prior to sampling.

Among 222 cases, 89 (40.1%) were positive for *S. pneumoniae* by NP culture, and an additional 38 cases were positive only by PCR (culture-negative); 83 (68.6%) of PCR+ cases were also culture-positive. No cases were positive for *S. pneumoniae* by blood culture, and only three cases had *S. pneumoniae* detected by whole blood PCR. Among 649 controls, 340 (52.4%) were NP culture-positive, and 76 additional controls were PCR+/culture-negative; 329 (81.2%) PCR+ controls were culture-positive. While not statistically different, cases were less likely to be pneumococcal culture-positive in the NP than controls (p = 0.07, adjusted for age and antibiotic use; Table 1).

The median pneumococcal colonization density among PCR-positives was lower for cases (42.1 x $10^3$ copies/mL, interquartile range (IQR): 2.1–223.3) than for controls (210.2 x $10^3$ copies/mL, IQR: 23.2–991.2; p <0.01). Three (1.4%) of 222 cases (2.5% of 121 PCR-positive cases) had colonization densities above a threshold associated with pneumococcal pneumonia identified in the all-site PERCH analysis (6.9 $\log_{10}$ copies/mL) [7], compared to 8 (1.2%) controls (2.0% of 406 PCR-positives) with colonization densities above this threshold (p = 0.84). We found no significant differences in pneumococcal colonization density among cases by site, age, gender, prior antibiotic use, or pneumonia severity (Table 2)**.** Controls with RTI had higher colonization density (505.8x $10^3$ copies/mL) than those without (109.5x $10^3$ copies/mL, p <0.0001) (S1 Table), and density was higher in controls from Nakhon Phanom (324.0 x $10^3$ copies/mL) than from Sa Kaeo (82.5 x $10^3$ copies/mL; p < 0.0001). Among cases (n = 30) and controls (n = 16) with co-detection of RSV, median colonization density was higher than in RSV-negative children but did not reach statistical significance. Median colonization density did not differ between CXR-positive cases and cases with normal CXRs (p = 0.18), or when restricting to cases with consolidation on CXR vs. cases with other infiltrate (p = 0.90). Of 6 cases with clinical characteristics consistent with possible bacterial pneumonia (CRP ≥40 mg/ L, alveolar consolidation, and no RSV co-infection), 4 were colonized with pneumococcus but the median density was low (7.2 x $10^3$ copies/mL).

Among 121 cases and 406 controls with pneumococcal NP colonization detected by PCR, 83 (68.5%) and 329 (81.0%), respectively, were also culture-positive, allowing for determination of serotype. Colonization isolates from an additional 24 children (N = 7 cases and N = 17 controls) who were NP culture positive for pneumococci but PCR-negative were also serotyped. The pneumococcal colonization serotype distribution was generally similar between cases and controls (Fig 1). The most common colonizing serotypes among cases were vaccine-types: Serotype 6B was detected in 23.1%, followed by 23F (17.8%), 19F (14.4%), 15B/C (10.0%), and 14 (8.9%); 70.0% of colonizing isolates were serotypes included in PCV10 and 76.7% were PCV13-type (Fig 2). Serotypes 6B, 23F and 19F were also the most common serotypes among controls (15.4%, 13.9% and 12.4%, respectively) with little difference between RTI and non-RTI controls; 61.8% were PCV10-type and 67.9% were PCV13-type. After adjusting for prior antibiotic use, age and study sites, there was no significant difference in presence of vaccine-type pneumococci comparing cases to controls for either PCV 10 (OR: 1.7, 95% CI 0.8–3.4, p = 0.12) or PCV 13 (OR: 1.7, 95% CI 0.8–3.4, p = 0.14). Although the pneumococcal colonization serotype distributions were slightly different between the two study sites, 7 of the 8 most frequent serotypes among cases were the same, and the top 6 serotypes among controls were the same (S1 and S2 Figs).

**Table 2. Pneumococcal nasopharyngeal PCR positivity and density among positives by case-control group.**

| Study groups | All cases N = 222 | | | | All controls N = 650 | | | | |
|---|---|---|---|---|---|---|---|---|---|
| | *S. pneumoniae* colonization prevalence | | | *S. pneumoniae* Colonization density | | *S. pneumoniae* colonization prevalence | | | *S. pneumoniae* Colonization density | |
| Characteristics | N | n (%) | p-value[a] | Median (IQR) ($10^3$ copies/ml) | p-value[b] | N | n (%) | p-value[a] | Median (IQR) ($10^3$ copies/ml) | p-value[b] |
| **Overall** | 222 | 121 (54.5)[c] | -- | 42.1 (2.1,223.3)[c] | -- | 650 | 406 (62.5) | -- | 210.2 (23.2,991.2) | -- |
| **Provincial site** | | | | | | | | | | |
| Nakhon Phanom | 145 | 76 (52.4) | 0.39 | 62.0 (2.1,576.5) | 0.12 | 429 | 274 (63.9) | 0.40 | 324.0 (36.1,1280.9) | < .0001 |
| Sa Kaeo | 77 | 45 (58.4) | | 19.0 (2.1,86.9) | | 221 | 132 (59.7) | | 82.5 (11.4,482.3) | |
| **Age** | | | | | | | | | | |
| 1–5 months | 38 | 14 (36.8) | **0.012** | 48.7 (0.7,1029.5) | 0.31 | 90 | 40 (44.4) | < .0001 | 502.4 (60.6,1536.9) | 0.30 |
| 6–11 months | 50 | 26 (52.0) | | 8.3 (1.73,126.0) | | 150 | 88 (58.7) | | 357.7 (51.9,915.2) | |
| 12–23 months | 68 | 35 (51.5) | | 84.9 (9.3,435.0) | | 212 | 124 (58.5) | | 177.0 (14.6,1189.2) | |
| 24–59 months | 66 | 46 (69.7) | | 45.9 (2.1,116.4) | | 198 | 154 (77.8) | | 185.0 (18.4,736.2) | |
| **Gender** | | | | | | | | | | |
| Male | 134 | 70 (52.2) | 0.49 | 42.2 (9.3,171. 9) | 0.43 | 330 | 209 (63.3) | 0.70 | 203.4 (27.9,1059.3) | 0.78 |
| Female | 88 | 51 (58.0) | | 18.4 (1.7,570.3) | | 320 | 197 (61.6) | | 222.6 (17.0,953.2) | |
| **Prior antibiotic use** | | | | | | | | | | |
| Any documented antibiotic pretreatment prior to specimen collection[d] | 94 | 50 (53.2) | 0.92 | 23.5 (1.9,124.0) | 0.21 | 5 | 1 (20.0) | 0.08 | 0.7 (—) | 0.12 |
| Parental report only | 51 | 28 (54.9) | | 38. 6 (2.0, 174.6) | | 51 | 35 (68.6) | | 330.7 (22.2, 1475.6) | |
| No evidence of antibiotic use | 77 | 43 (55.8) | | 63.9 (3.4, 614.8) | | 594 | 370 (62.3) | | 205.6 (23.7, 965.1) | |
| **NP culture positive for pneumococcus** | | | | | | | | | | |
| Yes | 89 | 83 (93.3) | < .0001 | 76.5 (10.2,593.0) | < .0001 | 340 | 329 (96.8) | < .0001 | 321.8 (64.0,1280.9) | < .0001 |
| No | 133 | 38 (28.6) | | 3.4 (0.5,64.2) | | 309 | 76 (24.6) | | 6.5 (0.9,177.1) | |
| **CRP ≥40 mg/L** | | | | | | | | | | |
| Yes | 37 | 19 (51.4) | 0.40 | 64.2 (4.8,96.7) | 0.88 | 0 | 0 (0) | - - - | - - - | - - - |
| No | 165 | 91 (55.2) | | 35.0 (1.9,399.7) | | 88[e] | 61 (69.3) | | 765.2 (133.6,2163.1) | |
| **RSV NPPCR positive** | | | | | | | | | | |
| Yes | 51 | 30 (58.8) | 0.41 | 80.8 (10.0,614.8) | 0.12 | 19 | 16 (84.2) | 0.066 | 380.4 (19.7,1078.0) | 0.67 |
| No | 171 | 91 (53.2) | | 26.2 (1.9,147.6) | | 631 | 390 (61.8) | | 205.6 (23.2,991.2) | |
| **Flu A/B NPPCR positive** | | | | | | | | | | |
| Yes | 7 | 3 (42.9) | 0.32 | 1.9 (0.4,6.5) | 0.08 | 0 | 0 (0.0) | - - - | - - - | - - - |
| No | 215 | 118 (54.9) | | 48.1 (2.1,399.7) | | 648 | 406 (62.7) | | 210.2 (23.2,991.2) | |
| **Any virus NPPCR positive** | | | | | | | | | | |
| Yes | 190 | 106 (55.8) | 0.44 | 40.4 (2.1,399.7) | 0.69 | 537 | 342 (63.7) | 0.07 | 217.9 (19.5,1041.7) | 0.85 |
| No | 31 | 15 (48.4) | | 86.9 (3.2,197.4) | | 113 | 64 (56.6) | | 185.5 (34.7,755.7) | |
| **Whole blood *lyt* A PCR positive** | | | | | | | | | | |

(*Continued*)

**Table 2.** (Continued)

| Study groups | All cases N = 222 | | | | | All controls N = 650 | | | | |
|---|---|---|---|---|---|---|---|---|---|---|
| | | *S. pneumoniae* colonization prevalence | | *S. pneumoniae* Colonization density | | | *S. pneumoniae* colonization prevalence | | *S. pneumoniae* Colonization density | |
| Characteristics | N | n (%) | p-value[a] | Median (IQR) ($10^3$ copies/ml) | p-value[b] | N | n (%) | p-value[a] | Median (IQR) ($10^3$ copies/ml) | p-value[b] |
| Yes | 3 | 2 (66.7) | 0.81 | 9.3 (0.3,18.4) | 0.19 | 5 | 4 (80.0) | 0.469 | 3,318.2 (2385.9,3863.2) | **0.005** |
| No | 219 | 119 (54.3) | | 42.3 (2.1,399.7) | | 609 | 387 (63.5) | | 203.4 (20.3,965.1) | |
| **Whole blood *lytA* PCR density $\geq$ 2.2 log10 copies/ml** | | | | | | | | | | |
| Yes | 0 | 0 (0.0) | ---- | ---- | ---- | 3 | 3 (100) | ---- | 3,017.5 (1754.3,3618.9) | **0.018** |
| No | 222 | 121 (54.5) | | 42.1 (2.1,223.4) | | 611 | 388 (63.5) | | 205.6 (20.5,967.8) | |
| **Observed cough** | | | | | | | | | | |
| Yes | 189 | 105 (55.6) | 0.74 | 38.6 (2.1,431.3) | 0.65 | 110 | 75 (68.2) | 0.17 | 481.9 (159.3,1569.6) | **0.0003** |
| No | 32 | 16 (50.0) | | 61.4 (3.8,83.7) | | 540 | 331 (61.3) | | 160.7 (17.8,919.6) | |
| **Hypoxemia[f]** | | | | | | | | | | |
| Yes | 54 | 29 (53.7) | 0.79 | 54.0 (5.6, 635.1) | 0.39 | - | - | | - | |
| No | 168 | 92 (54.8) | | 36.4 (1.9,184.7) | | - | - | | - | |
| **Chest X-ray (CXR)** | | | | | | | | | | |
| Positive | 99 | 59 (59.6) | 0.23 | 64.2 (2.9,582.8) | 0.18 | - | - | | - | |
| Negative | 96 | 50 (52.1) | | 15.2 (1.9,126.0) | | - | - | | - | |
| **CXR positive findings** | | | | | | | | | | |
| Any consolidation | 43 | 23 (53.5) | 0.32 | 76.5 (2.1,635.1) | 0.90 | - | - | | - | |
| Other infiltrate | 56 | 36 (64.3) | | 59.4 (3.8, 362.8) | | - | - | | - | |
| **Pneumonia severity** | | | | | | | | | | |
| Severe | 170 | 96 (56.5) | 0.27 | 48.1 (2.6,311.5) | 0.29 | - | - | | - | |
| Very Severe | 52 | 25 (48.1) | | 6.4 (1.7,171.9) | | - | - | | - | |
| **Died** | | | | | | | | | | |
| Yes | 9 | 1 (11.1) | **0.043** | 5.6 (—) | 0.56 | - | - | | - | |
| No | 208 | 118 (56.7) | | 42.2 (2.1,223.3) | | - | - | | - | |
| **Possible bacterial pneumonia[g]** | | | | | | | | | | |
| Yes | 7 | 4 (57.1) | 0.67 | 7.2 (1.2,49.8) | 0.28 | - | - | | - | |
| No | 204 | 112 (54.9) | | 40.4 (2.0,210.3) | | - | - | | - | |

IQR, Inter quartile range; PCR, Polymerase chain reaction; NP, Nasopharyngeal; CRP, C-reactive protein; RSV, Respiratory syncytial virus; NP PCR, Nasopharyngeal Polymerase Chain Reaction; CXR, Chest radiography; Flu A/B, Influenza A/B.

a. P-values obtained from logistic regression adjusted for age, comparing pneumococcal colonization prevalence by characteristic within study group (cases or controls).

b. P-values obtained from Kruskal-Wallis, comparing pneumococcal colonization density by characteristic within study group (cases or controls).

c. p = 0.12 comparing PCR+ in All cases vs All controls (logistic regression adjusting for age in months); p < 0.01 comparing median density among positives in All cases vs All controls (Kruskal-Wallis)

d. Presence of antibiotics by serum, antibiotics at the referral hospital, clinician report of antibiotics prior to specimen collection or antibiotics prior to NP specimen collection based on time of specimen collection and time of antibiotic administration. Only criterion applicable to controls is serum. P-values calculated excluding those with parental report only.

e. CRP was only assessed in a subset of controls, those who were positive for pneumococcus by whole-blood PCR, had RTI, or were in the top 25% for total NP/OP PCR pathogen load (across all pathogens tested for).

f. Hypoxemia defined as oxygen saturation < 92% on room air at admission or oxygen requirement (if no room air reading available).

g. Possible bacterial pneumonia defined as having CRP> 40 mg/L, alveolar consolidation, and RSV not detected by nasopharyngeal PCR nor induced sputum PCR (excludes children with unknown possible bacterial pneumonia status).

**Bolded p-values < 0.05.**

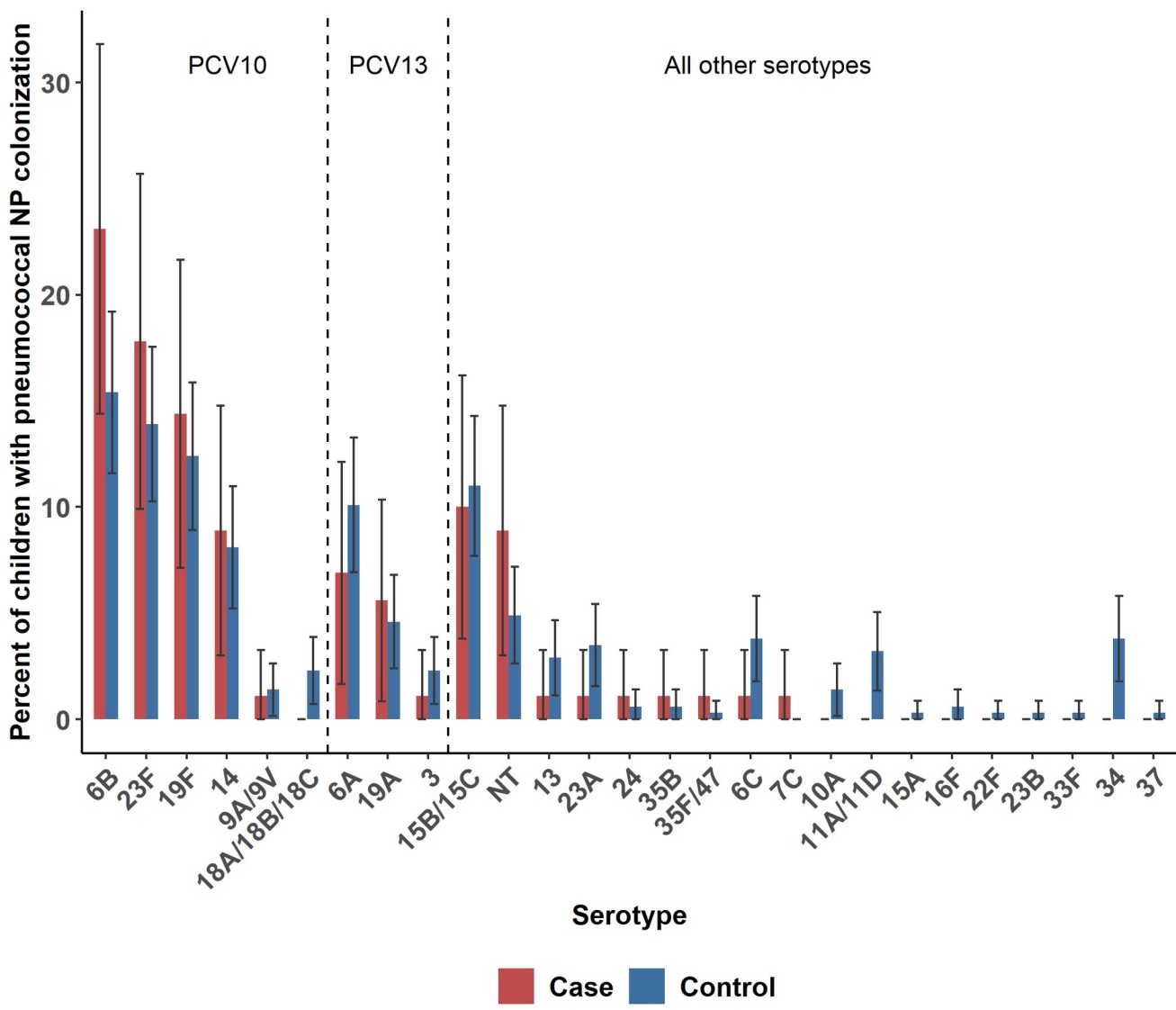

**Fig 1. Pneumococcal colonization serotype distribution among pneumonia cases and community controls in rural Thailand.** Bars represent the percent of cases or controls with the serotyped detected, restricted to children with NP culture serotyping data. Error bars are 95% Wald confidence intervals. PCV-10 serotypes: 1, 4, 5, 6B, 7F, 9V, 14, 18C, 19F, and 23F; PCV-13 serotypes: PCV-10 serotypes plus serotypes 3, 6A, and 19A. NT, non-typeable.

## Discussion

In children aged <5 years in the PERCH study in Thailand, we found that pneumococcal colonization was common in both children hospitalized with severe pneumonia and community controls. Antibiotic exposure before specimen collection was common among cases and may have contributed to the lower colonization density compared to controls [34]. Among children with severe pneumonia, colonization density was not elevated in children with characteristics of bacterial pneumonia compared to those without such characteristics. Among cases, 70.0%-76.7% of colonizing pneumococcal serotypes were PCV-types, exceeding slightly the percentage PCV-type in controls (61.8%-67.9%). These findings add to the limited knowledge on pneumococcal colonization and density in Southeast Asia and provide important data to advance discussions about potential future introduction of PCV in Thailand.

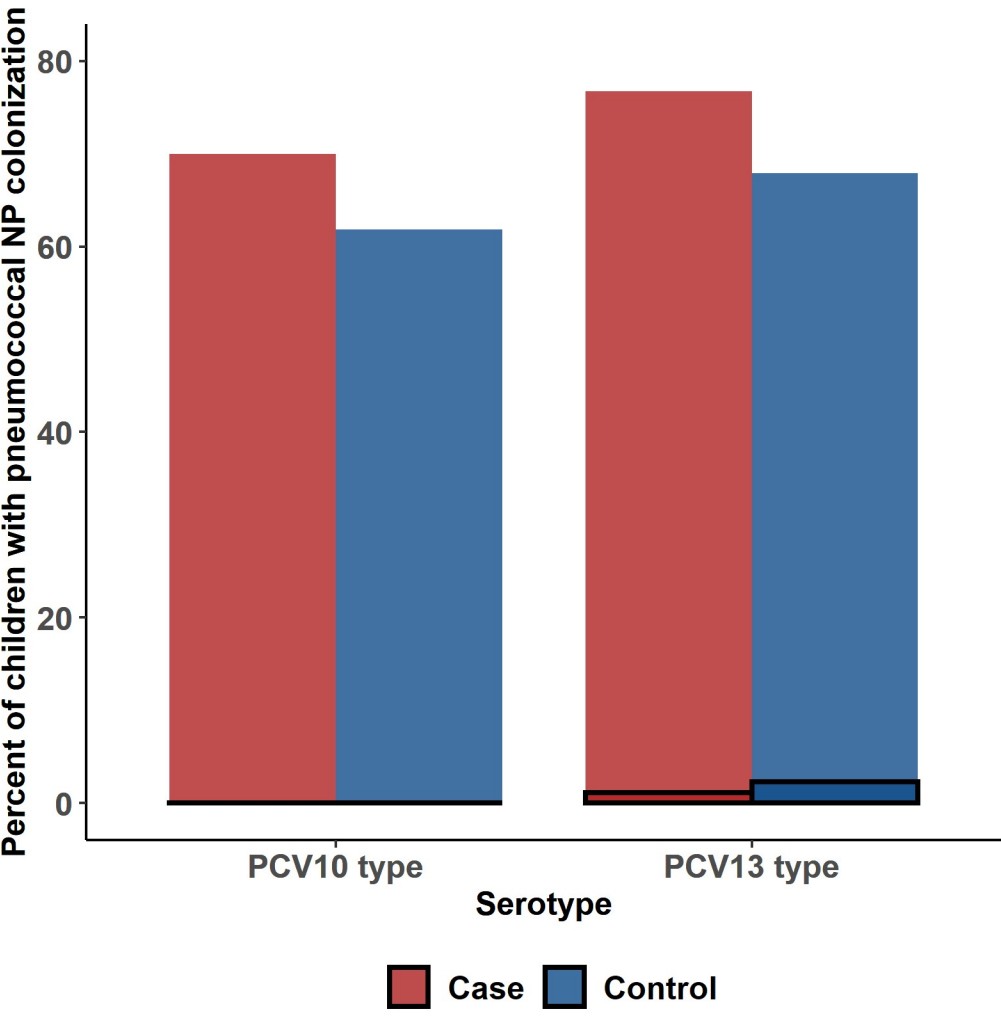

Black outline indicates serotype 3

**Fig 2. Pneumococcal colonization serotype distribution among pneumonia cases and community controls by pneumococcal conjugate vaccine type**\*. \*Pneumococcal conjugate vaccine (PCV)-10 serotypes: 1, 4, 5, 6B, 7F, 9V, 14, 18C, 19F, and 23F PCV-13 serotypes: PCV-10 serotypes plus serotypes 3, 6A, and 19A (ST 3 displayed separately). Restricted to children with NP culture serotyping data.

Previous colonization studies in Thailand have been conducted in selected populations, such as nursery and school attendees [13], infants in refugee camps [14], and hospitalized pneumonia patients and outpatients with influenza-like illness [15]. But such populations are unlikely to represent the true colonization ecology of pneumococcus in the community. Our inclusion of randomly selected community-based controls offers a more representative estimate of community colonization prevalence, which can be important for considering and monitoring the impact of PCV introduction.

Pneumococcal colonization prevalence in our controls (62.5%) was similar to what has been reported among healthy children in low-income countries in Africa and Asia before PCV introduction (64.8% [95 CI 49.8–76.1%]), based on a 2014 meta-analysis [6]. However, that analysis found high heterogeneity by country, with prevalence in several countries (all African) over 80%; the colonization prevalence from our study was somewhat higher than estimates for

other middle-income countries (47.8%, 95% CI 44.7–50.8%) [6]. Our controls included 38.5% with RTI, who had a higher colonization prevalence (68.8%) than did the non-RTI controls (58.5%). However, the colonization prevalence among non-RTI controls was still higher than the prevalence reported in other Southeast Asian countries (50.30% in healthy community children aged <5 years in Vietnam [10], 43% - 46% among healthy children in Indonesia [17, 18], 35% from nasal swab specimens in daycare centers in Malaysia [19], and 14.3% among children aged 5–8 weeks and 55.8% among those aged 12–23 months in Laos PDR [21]. These differences may in part be due to variations in specimen collection and testing methods. We tested two specimens (NP and OP), which may have increased detection sensitivity compared to studies that collected specimens from OP or nares only [27]. Some also used culture methods only, but prevalence in our study based only on NP culture (52.4% in controls) was still higher than prevalence reported from other Southeast Asian countries.

While pneumococcal colonization prevalence among our Thai controls was similar to or higher than prevalence reported from many low- or middle-income countries before the PERCH study, colonization prevalence was lower than that at other PERCH sites, which used the same standardized approaches to study enrollment, specimen collection, and testing as at the Thailand site [7]. Therefore, we are confident that differences in colonization prevalence between Thailand and other PERCH study sites likely represent real differences rather than differences attributable to variations in participant selection, specimen collection, or testing procedures. Such differences in colonization could be due to different exposure factors, such as population density, crowded living conditions, regular interaction with many different people, or exposure to other young children such as siblings or via day care [35, 36]. These factors differ by cultural practices and urban vs. rural settings, both between and within countries. Prevalence is also a function of the age of acquisition of carriage, which tends to be earlier in low- compared to high-income countries [3], meaning that middle-income countries like Thailand are likely in between. The lower colonization prevalence among the Thailand cases compared to controls could be attributed to prior antibiotic use, since 42.0% of cases received prior antibiotics compared to 0.8% of controls; however, colonization prevalence in cases did not differ significantly between those with and without prior antibiotic exposure.

Although pneumococcal colonization does not indicate disease, high density of colonization has been associated with confirmed pneumococcal disease in the PERCH study, and a density threshold of >6.9 $\log_{10}$ copies/mL distinguished microbiologically-confirmed pneumococcal cases from community controls (64% sensitivity, 92% specificity) [7]. There were no PERCH cases with laboratory-confirmed invasive pneumococcal disease (IPD) at the Thai site, which may explain the lower colonization density among cases compared to controls, even among cases with alveolar consolidation on CXR. However, higher density in suspected or confirmed pneumococcal cases has been seen elsewhere, including in Southeast Asia. Two studies in Vietnam found pneumococcal NP density among children with radiographically-confirmed pneumonia was significantly higher than density among healthy controls and children with acute lower respiratory tract infection [9, 10]. Studies in South Africa and Mozambique also found higher colonization density among children with IPD compared to controls [8, 11]. The colonization density among PERCH cases in Thailand was lower than the density among cases without microbiologically-confirmed pneumococcal disease at other PERCH sites, suggesting that pneumococcal disease may be less common among those presenting to hospital with severe pneumonia in Thailand compared to the other sites. Receipt of antibiotics prior to specimen collection may also have reduced median colonization density, as antibiotics have been associated with lower colonization density in all PERCH sites [7].

Our study describes the serotype distribution of colonizing pneumococcal isolates in Thailand. We found that a relatively high percentage of colonizing isolates from pneumonia cases

and controls were serotypes included in PCV10 (70.0% and 61.8%, respectively), and PCV13 (76.7% and 67.9%, respectively). Serotype distributions among cases and controls were similar, with only small differences in rank order of the top 8 serotypes. Certain serotypes (3, 13, 23A, 6C, 34, 11A/11D, and 18A/18B/18C) were found less frequently in cases than representative community controls, but each of these seroptyes represented <5% of all colonizing serotypes. Enrollment of randomly selected community controls provides an unbiased description of the circulating pneumococcal serotypes. The percentage of colonizing isolates that were PCV-type among cases was lower than previously described for blood isolates from hospitalized children aged <5 years (PCV10 = 74%, PCV13 = 92%) from these same provinces [37], as expected since PCVs include the more invasive serotypes [38]. The proportion of colonizing isolates in our study that were PCV13-type was similar to that from studies in Malaysia (69.5%) [19] and Cambodia (70.2%) [39] but higher than results from Indonesia (56% and 45%) [17, 18]. Previous studies in Thailand have described serotype distribution of colonizing pneumococci among Karen-infants in displaced persons camps [14] and among children with influenza-like illness (ILI) and pneumonia [15]. Although not necessarily representative of colonization in the general community, the top six pneumococcal serotypes from these studies were the same as the top six in PERCH Thailand cases and controls with the exception of 15B, which was not top six in the study of children with ILI and pneumonia [15]. Although Thailand does not include PCV in the National Immunization Program, our data can contribute to future policy discussions about potential introduction. A new conjugate vaccine in Phase II trials includes the same serotypes as PCV13 minus serotype 3 (detected in only 1% of cases); if this vaccine proves to be more affordable than PCV13, it could have important implications for vaccine policy considerations [33]. A recent study concluded that when considering herd-effect, both PCV10 and PCV13 would be cost-effective in Thailand [40]. In the next 3–5 years, Thailand's Advisory Committee on Immunization Practices (ACIP), a sub-steering committee of Thai National Vaccine Committee, will consider adding PCV to Thailand's Expanded Program on Immunization [Personal communication].

Our study had some limitations. First, we found no blood culture-confirmed *S. pneumoniae* among PERCH cases in Thailand, which meant we had no confirmed pneumococcal pneumonia cases as true positives. Second, both study provinces were relatively rural settings, so our findings might not be generalizable to urban settings like Bangkok. Third, all case-control studies are subject to potential selection bias related to control selection. We believe that selection bias was minimized by our approach to control selection; community controls were randomly selected from comprehensive lists of children aged 1–59 months drawn from health services registries, which include virtually all children in each province, and were enrolled year round and frequency matched to the age-group distribution of the cases on a monthly basis. Therefore, our controls represented an unbiased community comparison group in terms of pneumococcal colonization prevalence [25]. Fourth, we did not collect data on the dose and duration of antibiotic exposure prior to enrollment, so we were unable to fully account for the effect of antibiotic use on pneumococcal colonization prevalence and densities. Fifth, serotype data were unavailable for 38 (31.4%) of the 121 pneumococcal PCR-positive cases and 76 (18.7%) of 406 PCR-positive controls who had negative cultures. It is possible that the serotype distribution of PCR-positive/culture-negative samples differed from that of culture-positive samples, which would bias our serotype distribution estimates. Finally, nasopharyngeal isolates from 16.7% of cases and 26.3% of controls were not fully sub-typed for serogroups 6, 15, 9, 11, and 18. For serogroup 6, we estimated the distribution of isolates that were sub-types 6A and 6B by apportioning the unresolved serogroup 6 isolates according to the distribution of the isolates for which we were able to resolve the sub-type; this approach assumes that isolates with

unresolved sub-typing had the same sub-type distribution as the fully typed isolates. We had insufficient data to apportion the other unresolved serogroups.

Our two-year, highly standardized study, provides estimates of pneumococcal colonization prevalence, density, and serotype distribution among children aged <5 years hospitalized with severe or very severe pneumonia in Thailand and in community controls. These findings provide important data to inform discussions about PCV introduction in Thailand and offer baseline data on prevalence and serotype distribution that could be monitored after vaccine introduction to evaluate impact. Additional studies could evaluate the potential association of pneumococcal pneumonia and colonization density among young children in Thailand with a sample that includes confirmed cases of pneumococcal disease. Although IPD has been documented to occur commonly in Thailand, including in the PERCH study provinces [41], no IPD cases were detected during the PERCH Study [42]. Additional studies could evaluate the potential association of pneumococcal pneumonia and colonization density among young children in Thailand with a sample that includes confirmed cases of pneumococcal disease.

## Disclaimer

The findings and conclusions in this report are those of the authors and do not necessarily represent the official position of the Centers for Disease Control and Prevention, the U.S. Department of Health and Human Services, or the U.S. government.

## Supporting information

**S1 Table. Pneumococcal nasopharyngeal PCR positivity and density among controls by acute respiratory illness symptoms.** RTI, Respiratory Tract Illness; IQR, Inter quartile range; PCR, Polymerase chain reaction; NP, Nasopharyngeal; CRP, C-reactive protein; RSV, Respiratory syncytial virus; NP PCR, Nasopharyngeal Polymerase Chain Reaction; Flu A/B, Influenza A/B. a. Comparison of colonization prevalence: p = 0.01 in RTI controls vs Non-RTI controls. Comparison of median density: p < .0001 in RTI controls vs Non-RTI controls; p < .0001 in All cases vs RTI controls; p < 0.01 in All cases vs Non-RTI controls. b. Presence of antibiotics by serum. P-values calculated excluding those with parental report only. P-values obtained from logistic regression adjusted for age (pneumococcal colonization prevalence) and Kruskal-Wallis (pneumococcal density). **Bolded p-values < 0.05.**
(PDF)

**S1 Fig. Pneumococcal colonization serotype distribution among pneumonia cases and community controls in rural Thailand: Nakhon Phanom.** Bars represent the percent of cases or controls with the serotyped detected, restricted to children with NP culture serotyping data. Error bars are 95% Wald confidence intervals. PCV-10 serotypes: 1, 4, 5, 6B, 7F, 9V, 14, 18C, 19F, and 23F; PCV-13 serotypes: PCV-10 serotypes plus serotypes 3, 6A, and 19A.
(PDF)

**S2 Fig. Pneumococcal colonization serotype distribution among pneumonia cases and community controls in rural Thailand: Sa Kaeo.** Bars represent the percent of cases or controls with the serotyped detected, restricted to children with NP culture serotyping data. Error bars are 95% Wald confidence intervals. PCV-10 serotypes: 1, 4, 5, 6B, 7F, 9V, 14, 18C, 19F, and 23F; PCV-13 serotypes: PCV-10 serotypes plus serotypes 3, 6A, and 19A.
(PDF)

**S1 Data.**
(PDF)

**S1 File.**
(PDF)

## Acknowledgments

We would like to thank Somyote Srijaranai, Tussanee Amorninthapichet, Peera areerat, Somchai Chuananon, Anusak Kerdsin, Sirirat Makprasert, Possawat Jorakate, Anchalee Jatapai, Patranuch Sapchookul, Prasong Srisaengchai, and Sununta Henchaichon for their contribution to this project. We sincerely thank Supannee Promthet for the dedication to the manuscript development. We offer sincere thanks to the children and families who participated in this study. We acknowledge the work of all PERCH Contributors who were involved in data collection at the local sites and central laboratories, members of the PERCH Chest Radiograph Reading Panel, the laboratory director David R. Murdoch (University of Otago), and Trevor Anderson from Canterbury Health Laboratories. We also acknowledge the substantial contributions of members of the PERCH Study Group.

## Author Contributions

**Conceptualization:** Barameht Piralam, Christine Prosperi, Maria Deloria Knoll, Susan A. Maloney, Katherine L. O'Brien, Henry C. Baggett.

**Data curation:** Barameht Piralam, Christine Prosperi, Somsak Thamthitiwat, Charatdao Bunthi, Pongpun Sawatwong, Ornuma Sangwichian, Melissa M. Higdon, Nora L. Watson, Maria Deloria Knoll, Wantana Paveenkittiporn, Julia Rhodes, Katherine L. O'Brien.

**Formal analysis:** Barameht Piralam, Christine Prosperi, Melissa M. Higdon, Nora L. Watson, Maria Deloria Knoll, Henry C. Baggett.

**Funding acquisition:** Maria Deloria Knoll, Pasakorn Akarasewi, Susan A. Maloney, Katherine L. O'Brien, Henry C. Baggett.

**Investigation:** Barameht Piralam, Christine Prosperi, Somsak Thamthitiwat, Charatdao Bunthi, Pongpun Sawatwong, Ornuma Sangwichian, Melissa M. Higdon, Maria Deloria Knoll, Wantana Paveenkittiporn, Chuwattana Chara, Pasakorn Akarasewi, Julia Rhodes, Susan A. Maloney, Katherine L. O'Brien, Henry C. Baggett.

**Methodology:** Christine Prosperi, Somsak Thamthitiwat, Charatdao Bunthi, Pongpun Sawatwong, Melissa M. Higdon, Nora L. Watson, Maria Deloria Knoll, Wantana Paveenkittiporn, Julia Rhodes, Katherine L. O'Brien, Henry C. Baggett.

**Project administration:** Barameht Piralam, Christine Prosperi, Somsak Thamthitiwat, Charatdao Bunthi, Pongpun Sawatwong, Melissa M. Higdon, Maria Deloria Knoll, Chuwattana Chara, Pasakorn Akarasewi, Susan A. Maloney, Katherine L. O'Brien, Henry C. Baggett.

**Resources:** Christine Prosperi, Maria Deloria Knoll, Pasakorn Akarasewi, Katherine L. O'Brien, Henry C. Baggett.

**Software:** Nora L. Watson.

**Supervision:** Barameht Piralam, Christine Prosperi, Somsak Thamthitiwat, Charatdao Bunthi, Maria Deloria Knoll, Wantana Paveenkittiporn, Chuwattana Chara, Cameron P. Hurst, Pasakorn Akarasewi, Julia Rhodes, Susan A. Maloney, Katherine L. O'Brien, Henry C. Baggett.

**Validation:** Barameht Piralam, Christine Prosperi, Somsak Thamthitiwat, Charatdao Bunthi, Pongpun Sawatwong, Ornuma Sangwichian, Melissa M. Higdon, Nora L. Watson, Wantana Paveenkittiporn, Julia Rhodes.

**Visualization:** Christine Prosperi, Melissa M. Higdon, Nora L. Watson.

**Writing – original draft:** Barameht Piralam, Christine Prosperi, Melissa M. Higdon, Maria Deloria Knoll, Henry C. Baggett.

**Writing – review & editing:** Barameht Piralam, Christine Prosperi, Somsak Thamthitiwat, Charatdao Bunthi, Pongpun Sawatwong, Ornuma Sangwichian, Melissa M. Higdon, Maria Deloria Knoll, Wantana Paveenkittiporn, Chuwattana Chara, Cameron P. Hurst, Pasakorn Akarasewi, Julia Rhodes, Susan A. Maloney, Katherine L. O'Brien, Henry C. Baggett.

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
