## [Decision Letter · Decision Letter 0]

2 Dec 2019

PONE-D-19-29663

Pneumococcal colonization prevalence and density among Thai children with severe pneumonia and community controls

PLOS ONE

Dear Mr. Piralam,

Thank you for submitting your manuscript to PLOS ONE. After careful consideration, we feel that it has merit but does not fully meet PLOS ONE’s publication criteria as it currently stands. Therefore, we invite you to submit a revised version of the manuscript that addresses the points raised during the review process.

The manuscript has been assessed by two reviewers; the comments are available below. The reviewers have raised a number of concerns about the clarity in presentation of the work and the data, they recommend revisions to improve the clarity in presentation and writing and to provide a fuller outline of the methodology and main results. Please carefully revise the manuscript to address all the points raised by the reviewers.

We would appreciate receiving your revised manuscript by Jan 16 2020 11:59PM. To enhance the reproducibility of your results, we recommend that if applicable you deposit your laboratory protocols in protocols.io, where a protocol can be assigned its own identifier (DOI) such that it can be cited independently in the future. For instructions see: http://journals.plos.org/plosone/s/submission-guidelines#loc-laboratory-protocols

We look forward to receiving your revised manuscript.

Kind regards,

Jose Melo-Cristino, M.D., Ph.D.

Academic Editor

PLOS ONE

Journal Requirements:

4. Thank you for stating the following in the Financial Disclosure section:

'This work was supported by the Bill & Melinda Gates Foundation (https://www.gatesfoundation.org/) grant 48968 to the International Vaccine Access Center, Department of International Health, Johns Hopkins Bloomberg School of Public Health, for the PERCH study.

Representatives from the Bill & Melinda Gates Foundation participated in site selection and in Pneumonia Methods Working Group meetings, which informed the study design. They had no role in the data collection, data analysis, data interpretation, or writing of the report.'

We note that one or more of the authors are employed by a commercial company: The Emmes Company

Reviewers' comments:

Reviewer's Responses to Questions

**Comments to the Author**

1. Is the manuscript technically sound, and do the data support the conclusions?

Reviewer #1: Yes

Reviewer #2: Yes

2. Has the statistical analysis been performed appropriately and rigorously? 

Reviewer #1: Yes

Reviewer #2: Yes

3. Have the authors made all data underlying the findings in their manuscript fully available?

Reviewer #1: No

Reviewer #2: Yes

4. Is the manuscript presented in an intelligible fashion and written in standard English?

Reviewer #1: Yes

Reviewer #2: Yes

5. Review Comments to the Author

Reviewer #1: This manuscript examines pneumococcal colonisation prevalence and density among Thai children with severe pneumonia and community controls.

This is a well written paper which includes data from the PERCH (pneumonia) aetiology study.

My comments are minor only:

Page 4, line 89: add "usually" to "pneumococcal colonization is usually much higher..."

Page 5, line 107: there have been 2 papers from Laos published in 2019 on pneumococcal community carriage and risk factors for high median density carriage- need to add. Vaccine (Satzke et al) and Plos One (Dunne et al) papers

PCV vaccination status of cases and controls needs to be added as PCV is available in the private market in Thailand with ~30% uptake. Analysis in Table 2 needs to have density by vaccination status

pg 8, stats analysis: add how the data were managed, and how it were summarised before analysed.

Table 2 is difficult to read with regard to the p values and which comparisons they are making eg. the row with Provincial site has 4 p values but it is unclear what comparisons they are referring to- is it the above rows or the rows below? Table needs to be reformatted for clarity.

Table 2

Although there was a lack of statistical significance between many of the study groups with regard to density, the median and IQR differences within the subgroups suggest a there is a difference- just that there is insufficient power to detect it eg median density in RSV positive was 80 (IQR10-614) vs RSV neg 26 (IQR 1.9-147). If two histograms were plotted on one graph for each subgroup, the difference maybe more obvious, although not statistically significant. Ditto for antibiotics vs no antibiotic; CRP; Flu A/B; and al the clinical study groups. Suggest plotting this data as overlapping histograms for each subgroup.

Discussion

pg 19: line 320: please add a reference

Line 339: typo-remove %

Line 343: also compare with Laos community carriage results

Reviewer #2: Piralam and coll. assessed pneumococcal nasopharyngeal (NP) colonization in hospitalized children with pneumonia and in controls in Thailand. They also assessed pneumococcal colonization density among young children hospitalized with pneumonia and the association with characteristics of pneumonia, and distribution of serotypes. They used the data of the large multicentre PERCH study.

224 pneumonia cases and 659 community controls were included in the 2 study sites; both located in Thailand. The results showed a lower proportion of NP colonization and a lower median colonization density in cases compared to controls. No statistical differences were found in sub-groups analyses for colonization density.

Nowadays, PCR on NP samples cannot be used as a diagnostic tool as it has an uncertain value for individual diagnosis. Association between pneumococcal NP colonization density and pneumonia is, to my opinion, a topic of great interest. As a lot of asymptomatic controls have a NP colonization; a high amount of S. pneumoniae might be associated with pneumonia (as the authors mentioned in Introduction section l.93 réf 7-11).

The presented results are unexpected and hypotheses to explain them should be clearly formulated. Overall analyses of the PERCH study showed an association between pneumococcal colonisation density and microbiologically-confirmed pneumonia and the results in the Thai sub-group are opposite (réf 7). Then, potential bias should be carefully raised. I think that the manuscript can easily be improved (especially Results and Discussion sections). Be careful with typographical errors.

Major comments

1. Table 2 has to be improved, particularly to help the reader to understand quickly the Table. It is unclear. For example, in each line of the study characteristics, there are several p-value (for example: age P=0.012, P=0.31 etc.). It is not clear at all what mean these p-values and what is compared in this Table. The authors may add a new column for p-value and precise in header information what is compared. The authors may also perhaps split the Table in 2 Tables as some characteristics were not assessed in Controls. I particularly don’t understand to what refer p-values for characteristics not measured in controls (such as hypoxemia).

2. The authors may add some characteristics in Table 1 (or appendix …) in order to better describe the studied population (ex: for cases % detection of pneumococcus in blood by PCR, symptoms, etc).

3. The text of the results section is quite difficult to read with numerous imprecisions (see minor comments). You may try to clarify and focus on the interesting results.

4. Figure 1 is not understandable. Some serotypes are missing on the x-axis. The authors do not mention if some proportions are different between cases and controls. What mean NT?

5. How was collected the variable “prior antibiotic exposure”? If the quite low proportion of NP carriage of pneumococcus in cases might be explained by a higher proportion of antibiotic intake in cases, then the proportion of NP carriage should be higher in the group without previous antibiotic intake compared to the group without antibiotic intake. What hypotheses can be formulated to explain this?

6. Discussion p. 22 l.374 and after. Did you find other studies showing a higher pneumococcal density in healthy controls compared to pneumonia cases? I think that you have to carefully mention potential alternative explanations to explain this unexpected result: were operators (clinicians who collect the swab on subjects) the same between the hospital study site and community? If controls may have RTI, is it possible that the sensitivity of the swab was higher/increased in controls compared to the sensitivity in pneumonia cases (in the case of a higher amount of mucus, etc)?

7. I think that you have to clearly mention the proportion of cases and controls vaccinated with PCV. If no cases and controls were vaccinated because PCV was not in the national immunization program during the stud, you should clearly mention it.

8. The authors have to clearly present the potential bias. What about a selection bias in the study if the controls are all included in a same population with a high prevalence of NP carriage of S. pneumoniae?

Minor comments

1. P.3 l.67 103 and not 103

2. The authors precise that they performed binary logistic regression adjusted for age; as the study was frequency matched for age. You mean that the study was not paired matched, and then, that it was not possible to perform a conditional logistic regression?

3. P. 17, l.261 I think that it is better to mention IQR with a “-“ and not “,” (for example 2.1-222.3)

4. P.17, l.262 is it possible to add the p-value between proportion of cases and control above the threshold identified in the overall PERCH analysis?

5. P.17, l. 264 mL instead of ml

6. P.17, l. 266 I don’t understand to what the p-value (p=0.31) refers? It may be better to mention “The density is highest […] even if not significant” or mention the p-values after the sentence “none of these differences reached statistical significance”

7. P.17, l. 274 Only 3 cases had pneumococcus detected by PCR in whole blood? You may mention in the population description

8. P. 17, l. 275 Do you mean “were colonized in NP” ?

9. P. 17, l. 278 Do you mean “in whole blood by PCR”?

10. P. 18, l. 286 “with NP colonization”

11. Figures 1 & 2 the caption of y-axis “percent of children” is not clear. You might mention “percent of children with pneumococcal NP colonization”

12. P. 18, l. 299 I think that it is better to mention 95% CI with a “-“ and not “,”

13. P. 19, l. 301 “7 of the 8 most frequent serotypes” instead of “the top 8 serotypes”

14. P. 20, l. 339 remove “%”

15. P. 24, l. 424 “though …. unresolved serotypes”. This sentence is difficult to understand, you may rewrite it. I don’t know the meaning of the word “apporportioned”. Do you mean

6. PLOS authors have the option to publish the peer review history of their article (what does this mean?). If published, this will include your full peer review and any attached files.

Reviewer #1: No

Reviewer #2: Yes: Cédric Dananché

---

## [Author Response · Author response to Decision Letter 0]

27 Jan 2020

Please find the response to both of major and minor comments from reviewers as details as below;

Major comments 

1. Table 2 has to be improved, particularly to help the reader to understand quickly the Table. It is unclear. For example, in each line of the study characteristics, there are several p-value (for example: age P=0.012, P=0.31 etc.). It is not clear at all what mean these p-values and what is compared in this Table. The authors may add a new column for p-value and precise in header information what is compared. The authors may also perhaps split the Table in 2 Tables as some characteristics were not assessed in Controls. I particularly don’t understand to what refer p-values for characteristics not measured in controls (such as hypoxemia).

Response: The p-values in the table are comparing S. pneumoniae colonization prevalence and density by characteristics within each study group (cases and controls). There are no comparisons between cases and controls displayed in the table. We have revised the structure to shift the p-values from the rows to dedicated columns. We have also added a footnote explaining the p-values. The revised table 2 was placed in the manuscript but the changes to the table are not tracked

2. The authors may add some characteristics in Table 1 (or appendix …) in order to better describe the studied population (ex: for cases % detection of pneumococcus in blood by PCR, symptoms, etc). 

Response: We appreciate the suggestion to add more information on characteristics of the cases and controls in table 1. Many of the important characteristics are included in table 2 (number with pneumococcus detected in blood, number with cough, deaths, etc), although we recognize that the information is stratified by pneumococcus detection in the NP/OP. We are willing to add some of these characteristics to table 1 but worry it might look a little redundant. Also, a full description of all cases and controls in Thailand is pending publication right now. As soon as that paper is accepted by the journal, we will add a reference to that paper into our paper under review at PLoS One. If the editor would like us to add further details to table 1, we can do so quickly. 

3. The text of the results section is quite difficult to read with numerous imprecisions (see minor comments). You may try to clarify and focus on the interesting results.

Response: Thank you for the comment. We have revised the results section in response to the reviewer’s specific comments below. In the process, we also made some additional adjustments for clarity, including shortening sections that were not critical to the paper’s main message. 

4. Figure 1 is not understandable. Some serotypes are missing on the x-axis. The authors do not mention if some proportions are different between cases and controls. What mean NT?

Response: Some serotypes were dropped when exporting the figure. We have revised the figures to correct this issue and include all serotypes (see revised Figure 1).

5. How was collected the variable “prior antibiotic exposure”? If the quite low proportion of NP carriage of pneumococcus in cases might be explained by a higher proportion of antibiotic intake in cases, then the proportion of NP carriage should be higher in the group without previous antibiotic intake compared to the group without antibiotic intake. What hypotheses can be formulated to explain this?

Response: “Any documented antibiotic pretreatment prior to specimen collection” was defined as presence of antibiotics by serum, antibiotics at the referral hospital, clinician report of antibiotics prior to specimen collection or antibiotics prior to NP specimen collection based on time of specimen collection and time of antibiotic administration. “Parental report only” defined as no documented evidence but report by parent or caregiver. In reviewing these definitions, we decided to make the comparisons cleaner by excluding from the analysis those with antibiotic pretreatment by parental report only. The revised p-values in table 2 and supplementary table 1 are only for the comparison between those with documented evidence of antibiotic pretreatment and those with no evidence of antibiotic pretreatment. Although the differences in NP/OP PCR positivity by antibiotic use were not statistically significant, we did observe differences in NP culture positivity by antibiotic use. 

6. Discussion p. 22 l.374 (Shifted to Page 24 line 382) and after. Did you find other studies showing a higher pneumococcal density in healthy controls compared to pneumonia cases? I think that you have to carefully mention potential alternative explanations to explain this unexpected result: were operators (clinicians who collect the swab on subjects) the same between the hospital study site and community? If controls may have RTI, is it possible that the sensitivity of the swab was higher/increased in controls compared to the sensitivity in pneumonia cases (in the case of a higher amount of mucus, etc)?

Response: Despite extensive review of the literature, we did not find other studies showing lower colonization density among pneumonia cases compared to controls. We speculated possible explanations in the discussion starting with line 375. Although statistically significant, given the wide range of densities seen, one could argue that the absolute difference in median density was not substantial. The study staff collecting the NP/OP swabs from cases were not the same as the staff collecting specimens from controls. However, as noted in the discussion, all clinical staff underwent rigorous training in standard specimen collection, with required demonstration of proficiency and periodic refresher trainings. Controls with RTI had significantly higher colonization density compared to controls without RTI, which supports the reviewer’s suggestion that increased mucous could increase pneumococcal yield.

7. I think that you have to clearly mention the proportion of cases and controls vaccinated with PCV. If no cases and controls were vaccinated because PCV was not in the national immunization program during the stud, you should clearly mention it.

Response: Only 2 cases were vaccinated, and no controls received one or more dose of PCV during the study period. We added this information in Page 20, line 304-305 

8. The authors have to clearly present the potential bias. What about a selection bias in the study if the controls are all included in a same population with a high prevalence of NP carriage of S. pneumoniae?

Response: We believe that potential selection bias was minimized by the approach to control selection. Community controls were randomly selected from comprehensive lists of children aged 1 - 59 months drawn from health services registries, which include virtually all children in each province. Controls were enrolled year round and frequency matched to the age-group distribution of the cases on a monthly basis. The control selection process is detailed in a separate paper that has been submitted for publication and will ultimately be referenced in this paper. 

Minor comments

1. Page 4, line 89 (shifted to Page 5 line 89): add "usually" to "pneumococcal colonization is usually much higher..."

Response: Revised as suggested

2. Page 5, line 107: there have been 2 papers from Laos published in 2019 on pneumococcal community carriage and risk factors for high median density carriage- need to add. Vaccine (Satzke et al) and Plos One (Dunne et al) papers

Response: Added 2 papers from Laos PDR.

3. PCV vaccination status of cases and controls needs to be added as PCV is available in the private market in Thailand with ~30% uptake. Analysis in Table 2 needs to have density by vaccination status

Response: The PCV vaccination status was added in Page 10 line 202. Given that there were only 2 cases who received PCV and no controls who received even one dose of PCV, we did not feel it necessary to include density by vaccination status in Table 2. 

4. Page 8 (shifted to Page 9 line 173-174), stats analysis: add how the data were managed, and how it were summarized before analyzed.

Response: We added the following sentence to the Statistical Analysis section and reference to Watson et al 2017 (Data Management and Data Quality in PERCH, a Large International Case-Control Study of Severe Childhood Pneumonia) which described the data management processes in PERCH. “Data were single-data entered into a centralized electronic data capture system and data cleaning occurred throughout the study (Emmes Corporation, Rockville, MD, USA).” 

5. Table 2 is difficult to read with regard to the p values and which comparisons they are making eg. the row with Provincial site has 4 p values but it is unclear what comparisons they are referring to- is it the above rows or the rows below? Table needs to be reformatted for clarity.

Response: The table was reformatted as the described above (major comment # 1)

6.Table 2 

Although there was a lack of statistical significance between many of the study groups with regard to density, the median and IQR differences within the subgroups suggest a there is a difference- just that there is insufficient power to detect it eg median density in RSV positive was 80 (IQR10-614) vs RSV neg 26 (IQR 1.9-147). If two histograms were plotted on one graph for each subgroup, the difference maybe more obvious, although not statistically significant. Ditto for antibiotics vs no antibiotic; CRP; Flu A/B; and al the clinical study groups. Suggest plotting this data as overlapping histograms for each subgroup.

Response: Thank you for the suggestion. We made histograms of colonization density by various characteristics (See separate attachment with histograms) to see if they revealed distinctions between groups. We see a lot of overlap in density between groups for each characteristic. We do not think the histograms add information that will be helpful to the reader and prefer not to include them in the manuscript. 

7. P.3 l.67 103 and not 103

Response: Revised to 103

8. The authors precise that they performed binary logistic regression adjusted for age; as the study was frequency matched for age. You mean that the study was not paired matched, and then, that it was not possible to perform a conditional logistic regression?

Response: Since the study was age-frequency matched, as opposed to individually matched, we didn’t think it is necessary to use conditional logistic regression. 

9. P. 17, l.261 (shifted to Page 18 line 265-266) I think that it is better to mention IQR with a “-“ and not “,” (for example 2.1-222.3)

Response: Revised

10. P.17, l.262 (shifted to Page 18 line 269) is it possible to add the p-value between proportion of cases and control above the threshold identified in the overall PERCH analysis?

Response: 3/222 (1.4%) of case and 8/650 (1.2%) of controls in Thailand had pneumococcal NP/OP PCR density above the threshold identified in the overall PERCH analysis (6.9 log10 copies/ml) (p=0.84). The p-value was added to the text.

11. P.17, l. 264 (shifted to Page 18 line 268) mL instead of ml

Response: Revised as suggested

12. P.17, l. 266 (shifted to Page 18 line 271) I don’t understand to what the p-value (p=0.31) refers? It may be better to mention “The density is highest […] even if not significant” or mention the p-values after the sentence “none of these differences reached statistical significance”

Response: This comment refers to the below sentence in the Results:

“Pneumococcal colonization density among cases was highest in those aged 12-23 months (p=0.31), and although slightly higher in males than females (p=0.43), in Nakhon Phanom vs. Sa Kaeo (p = 0.12), and in cases with severe vs. very severe pneumonia (p = 0.29), none of these differences reached statistical significance.” These p-values were obtained from a Kruskal-Wallis test, comparing pneumococcal colonization density by characteristic within study group (cases or controls). Upon further review, we have decided to revise the sentence to note the lack of statistically significant differences and remove the p-values.

13. P.17, l. 274 Only 3 cases had pneumococcus detected by PCR in whole blood? You may mention in the population description

Response: This is correct. We have added this detail to Page 18 line 258-259.

14. P. 17, l. 275 Do you mean “were colonized in NP” ?

Response: Yes, this sentence referred to NP colonization among cases and controls who were positive for pneumococcus by PCR in whole blood. Because there were so few cases and controls positive for pneumococcus in whole blood, we decided to delete this information from the text. The information remains in table 2, and we believe that this change simplifies the text. 

15. P. 17, l. 278 Do you mean “in whole blood by PCR”?

Response: Yes, but because there were so few cases and controls positive for pneumococcus in whole blood and the data are in table 2, we deleted the sentence from the text. 

16. P. 18, l. 286 (shifted to Page 20 line 292) “with NP colonization” 

Response: We added ‘NP’ to make sure this was clear. 

17. Figures 1 & 2 the caption of y-axis “percent of children” is not clear. You might mention “percent of children with pneumococcal NP colonization”

Response: It was changed to “Percent of children with pneumococcal NP colonization” as suggested 

18. P. 18, l. 299 (shifted to Page 20 line 304) I think that it is better to mention 95% CI with a “-“ and not “,”

Response: Revised

19. P. 19, l. 301 (shifted to Page 20 line 306) “7 of the 8 most frequent serotypes” instead of “the top 8 serotypes”

Response: Revised as suggested

20. Page 19: line 320 (shifted to Page 21 line 327): please add a reference

Response: We added a reference supporting the potential association between antibiotic use and lower colonization density. 

21. P. 20, 

Line 339 (shifted to Page 22 line 346): remove “%” 

Response: Removed

22. P. 22

Line 343 (Shifted to Page 22 line 351-352): also compare with Laos community carriage results

Response: Added a sentence to reference the recently published cross-sectional carriage study from Laos PDR

23. P. 24, l. 424 (Shifted to Page 26-27 line 431-437) “though …. unresolved serotypes”. This sentence is difficult to understand, you may rewrite it. I don’t know the meaning of the word “apporportioned”. Do you mean “apportioned”? What are the “unresolved serotypes”?

Response: The “unresolved serotypes” refers to isolates for which we could determine serogroup but not fully resolve the sub-type (e.g., we could identify that the isolate was serogroup 6 but not determine whether A or B). Serotypes 6A/6B by PCR could not be differentiated for the samples collected in 2013, so we applied the 6B:6A ratio from 2012 to the 2013 data. This calculation was done separately by site and case-control status which was described in the Statistical analysis, page 9 line 178-181. For the other serogroups we did not have sufficient data on the sub-types to perform this same calculation. We corrected the typographical error, with “apporportioned” revised to “apportioned”

---

## [Decision Letter · Decision Letter 1]

5 Feb 2020

PONE-D-19-29663R1

Pneumococcal colonization prevalence and density among Thai children with severe pneumonia and community controls

PLOS ONE

Dear Mr. Piralam,

Thank you for submitting your manuscript to PLOS ONE. After careful consideration, we feel that it has merit but does not fully meet PLOS ONE’s publication criteria as it currently stands. Therefore, we invite you to submit a revised version of the manuscript that addresses the points raised during the review process.

You have now addressed most of the issues raised by the reviewers and the paper already clarifies most of the points brought up. However, two outstanding points remain. These are questions 2 and 8 of the reviewer. Please add the reference of the accepted paper you allude to to the current submission or, if the paper was not yet accepted, please provide the requested details in table 1 and a more completed discussion of a potential selection bias and the steps you took to minimize it in the text.

We would appreciate receiving your revised manuscript by Mar 21 2020 11:59PM. To enhance the reproducibility of your results, we recommend that if applicable you deposit your laboratory protocols in protocols.io, where a protocol can be assigned its own identifier (DOI) such that it can be cited independently in the future. For instructions see: http://journals.plos.org/plosone/s/submission-guidelines#loc-laboratory-protocols

We look forward to receiving your revised manuscript.

Kind regards,

Jose Melo-Cristino, M.D., Ph.D.

Academic Editor

PLOS ONE

Reviewers' comments:

Reviewer's Responses to Questions

**Comments to the Author**

1. If the authors have adequately addressed your comments raised in a previous round of review and you feel that this manuscript is now acceptable for publication, you may indicate that here to bypass the “Comments to the Author” section, enter your conflict of interest statement in the “Confidential to Editor” section, and submit your "Accept" recommendation.

Reviewer #1: All comments have been addressed

2. Is the manuscript technically sound, and do the data support the conclusions?

Reviewer #1: Yes

3. Has the statistical analysis been performed appropriately and rigorously? 

Reviewer #1: Yes

4. Have the authors made all data underlying the findings in their manuscript fully available?

Reviewer #1: Yes

5. Is the manuscript presented in an intelligible fashion and written in standard English?

Reviewer #1: Yes

6. Review Comments to the Author

Reviewer #1: (No Response)

7. PLOS authors have the option to publish the peer review history of their article (what does this mean?). If published, this will include your full peer review and any attached files.

Reviewer #1: No

---

## [Author Response · Author response to Decision Letter 1]

14 Feb 2020

I would like to provide a point-by-point response to the two remaining requests related to our manuscript:

1. Major comment no. 2; the authors may add some characteristics in Table 1 (or appendix …) in order to better describe the studied population (ex: for cases % detection of pneumococcus in blood by PCR, symptoms, etc). 

Response: We have added key laboratory and clinical characteristics to table 1, which complements the information in table 2 (stratified by pneumococcal NP/OP positivity). These variables include sub-site, sex, pneumococcal positivity in the NP/OP by PCR and culture, pneumococcal positivity in whole blood, and clinical symptoms (cough, fever). Also, a full description of all cases and controls in Thailand is pending publication. If that paper will be accepted before publication of this paper, we will add a reference to that paper into our paper under review at PLoS One. 

2. Major comment no. 8; the authors have to clearly present the potential bias. What about a selection bias in the study if the controls are all included in a same population with a high prevalence of NP carriage of S. pneumoniae?

Response: We recognize that all case-control studies are subject to potential selection bias. To acknowledge this point, we added the following sentences to the limitations paragraph in the Discussion section (lines 412-419): Third, all case-control studies are subject to potential selection bias related to control selection. We believe that selection bias was minimized by our approach to control selection; community controls were randomly selected from comprehensive lists of children aged 1 - 59 months drawn from health services registries, which include virtually all children in each province, and were enrolled year round and frequency matched to the age-group distribution of the cases on a monthly basis. Therefore, our controls represented an unbiased community comparison group in terms of pneumococcal colonization prevalence.

---

## [Editor Report · Decision Letter 2]

9 Apr 2020

Pneumococcal colonization prevalence and density among Thai children with severe pneumonia and community controls

PONE-D-19-29663R2

Dear Dr. Piralam,

We are pleased to inform you that your manuscript has been judged scientifically suitable for publication and will be formally accepted for publication once it complies with all outstanding technical requirements.

With kind regards,

Jose Melo-Cristino, M.D., Ph.D.

Academic Editor

PLOS ONE
---

## [Editor Report · Acceptance letter]

14 Apr 2020

PONE-D-19-29663R2 

Pneumococcal colonization prevalence and density among Thai children with severe pneumonia and community controls 

Dear Dr. Piralam:

I am pleased to inform you that your manuscript has been deemed suitable for publication in PLOS ONE. Congratulations! Your manuscript is now with our production department. 

With kind regards,

on behalf of

Prof. Jose Melo-Cristino 

Academic Editor

PLOS ONE